# Parkinson’s Disease: From Genetics and Epigenetics to Treatment, a miRNA-Based Strategy

**DOI:** 10.3390/ijms24119547

**Published:** 2023-05-31

**Authors:** Elena Paccosi, Luca Proietti-De-Santis

**Affiliations:** Unit of Molecular Genetics of Aging, Department of Ecology and Biology (DEB), University of Tuscia, 01100 Viterbo, Italy

**Keywords:** Parkinson’s disease, genetics, epigenetics, exosomes, miRNA

## Abstract

Parkinson’s disease (PD) is one of the most common neurodegenerative disorders, characterized by an initial and progressive loss of dopaminergic neurons of the *substantia nigra pars compacta* via a potentially substantial contribution from protein aggregates, the Lewy bodies, mainly composed of α-Synuclein among other factors. Distinguishing symptoms of PD are bradykinesia, muscular rigidity, unstable posture and gait, hypokinetic movement disorder and resting tremor. Currently, there is no cure for PD, and palliative treatments, such as Levodopa administration, are directed to relieve the motor symptoms but induce severe side effects over time. Therefore, there is an urgency for discovering new drugs in order to design more effective therapeutic approaches. The evidence of epigenetic alterations, such as the dysregulation of different miRNAs that may stimulate many aspects of PD pathogenesis, opened a new scenario in the research for a successful treatment. Along this line, a promising strategy for PD treatment comes from the potential exploitation of modified exosomes, which can be loaded with bioactive molecules, such as therapeutic compounds and RNAs, and can allow their delivery to the appropriate location in the brain, overcoming the blood–brain barrier. In this regard, the transfer of miRNAs within Mesenchymal stem cell (MSC)-derived exosomes has yet to demonstrate successful results both in vitro and in vivo. This review, besides providing a systematic overview of both the genetic and epigenetic basis of the disease, aims to explore the exosomes/miRNAs network and its clinical potential for PD treatment.

## 1. Introduction

Parkinson’s disease (PD) is one of the most common neurodegenerative disorders, second only to Alzheimer’s disease (AD). Distinguishing symptoms of PD are bradykinesia, muscular rigidity, unstable posture and gait, hypokinetic movement disorder and resting tremor [1,2]. Moreover, non-motor features, such as dementia, depression and dysautonomia, have been described [3]. The overall motor disturbances of PD are referred to as Parkinsonism. Although Parkinsonism is mainly associated with PD, other diseases, such as AD- and PD-related disorders, share these same features [4]. Both sporadic and familial forms of PD are characterized by an initial and progressive loss of dopaminergic neurons (DA) of the *substantia nigra pars compacta* (*SNc*) via a potentially substantial contribution from protein aggregates, the Lewy bodies (LB), mainly composed, among other factors, of α-Synuclein (α-Syn) [3], a protein responsible of regulating the release of neurotransmitters from synaptic vesicles in the brain. LBs were initially thought to be the pathophysiologically relevant form of α-Syn, and this is why neuron loss was thought to be the first step in the neurodegeneration found in PD. However, today, the idea is that the synaptic dysfunction of still-existing nerve cells is the primary event in the pathophysiology of PD, and not the death of dopaminergic neurons [5]. Even if there is a general consensus on the leading role of the accumulation of misfolded α-Syn in the progression of PD, however, its pathogenic role is still poorly understood due to the interplay between a plethora of genetic/molecular factors and environmental conditions involved in α-Syn accumulation [6]. Among them, worth mentioning, are mitochondrial dysfunction, oxidative damage and neuroinflammation, as well as the dysregulation of signaling pathways implicated in cell survival, apoptosis and autophagy, all of them leading to the accumulation of misfolded α-Syn [7,8,9,10]. Furthermore, the α-Syn microenvironment may impact its conformational state by inducing α-Syn polymerization, fibrillation and propagation [11,12]. As a detrimental loop, the α-Syn oligomers induce mitochondrial dysfunction and trigger endoplasmic and oxidative stresses, neuroinflammation and inhibition of both autophagy and proteasomal activities [12]. In addition, in many cases, there are mitochondrial effects, which may not necessarily require or derive from the increases in α-Syn, and a wide variety of other rationales for oxidative processes as further potential causative factors that might derive via mechanisms alternative to α-Syn [13,14], even if α-Syn and related Lewy bodies seem to remain one of many contributing factors to the deterioration in Parkinson’s disease [15].

## 2. Parkinson’s Disease: Genetics and Epigenetics

### 2.1. Parkinson’s Genetics

For years, there has been a clear dissociation in driving mechanisms between the familial and, thus, more hereditary versions of Parkinsonism; nevertheless, the larger numbers of patients clearly have a wider variety of different factors at work, as explained above [6,7,8,9,10,13,14].

Various genes are implicated as monogenic causes of PD, or, for disorders with Parkinsonism, among the former, mutations in *SNCA* (or *PARK1*) and *LRRK2* (or *PARK8*) are associated with the autosomal dominant inheritance of PD, while mutations in *PRKN* (or *PARK2*), *PINK1* (or *PARK6*) and *DJ-1* (or *PARK7*) [16] are responsible for the forms inherited in an autosomal recessive manner.

Regarding the *SNCA* gene, the mechanisms by which its mutations lead to PD are barely known. To date, the accepted model is of a gain-of-function mechanism, able to induce the aberrant aggregation of the α-Syn protein and, consequently, lead to cell damage and neuronal death [17,18]. The first missense mutation in the *SNCA* gene (A53T in exon 4) was identified in 1997 in an Italian family showing autosomal dominant inheritance of PD [19]. Further studies of different PD clinical cases reported other missense mutations in *SNCA*: A30P [20], E46K [21], G51D [22], H50Q and A53E mutations [23]. These kinds of mutations are associated with early-onset PD and display typical motor symptoms, showing a good response to Levodopa (L-dopa) treatment. Anyway, recent sequence analyses of a wide panel of patients revealed that qualitative mutations such as point mutations in *SNCA* are not the most common cause of PD [24,25,26,27,28,29,30]. Indeed, a plethora of quantitative mutations such as duplication or triplication of wild-type *SNCA*, inherited in an autosomal dominant manner, have been found in many patients characterized by both typical motor and non-motor symptoms; it was also observed that gene triplication results in an earlier onset of disease as compared with its duplication [31,32,33,34]. Furthermore, it has been reported that, besides gene amplifications, increased gene expression due to variations in the promoter region of *SNCA* may also increase susceptibility for PD. In particular, it was shown that a microsatellite repeated sequence, termed NACP-Rep1, in cooperation with its binding protein, PARP-1, has a physiological function in the regulation of SNCA gene expression. NACP-Rep1 has two domains flanking the repeat able to enhance the expression of *SNCA*, whereas the repeat itself may act as a negative modulator of *SNCA.* Hypothetical polymorphisms at this microsatellite region, via their impact on SNCA, may represent major culprits contributing to the risk for PD [35].

Additionally, mutations in Leucine-Rich Repeat Kinase 2 gene (*LRRK2* or *PARK8*) are among the major causes of familial PD. *LRRK2* codes for a protein kinase [36,37] that owns at the C-terminal five functional domains, all of them being subjected to disease-producing mutations [38,39]. A recent work identified around 50 mutations in *LRRK2* related to both familial and sporadic forms of PD [40]. All these mutations confer an increased risk of developing PD even if, in some cases, their effect is modest due to their low penetrance. It was observed that all the mutations studied so far up-regulate kinase activity and increase the autophosphorylation of LRRK2. Indeed, G2019S, the most common and highly penetrant mutation in *LRRK2* and also the most common monofactorial cause of PD identified until now, increases the autophosphorylation of LRRK2 in a gain-of-function pathogenic mechanism [41]. Patients carrying some common LRRK2 mutations, such as G2019S, R1441C and Y1699C, all present the typical late-onset PD symptoms, including non-motor symptoms; Lewy body disease; and degeneration of DA neurons in the SNc with neurofibrillary tangles, abnormal tau deposits and neuronal intranuclear inclusions [37,42,43].

More recently, new genes have been recognized for their contribution to monogenic-dominant PD: Glucocerebrosidase (*GBA*), which encodes a lysosomal protein that degrades glucocerebroside [44], and Vacuolar Sorting Protein 35 (*VPS35*) [45,46], the chaperone of the Hsp40 family *DNAJC13* [47], *CHCHD2* [48], *TMEM230* [49] and *RIC3* [50]. However, definitive confirmation of the pathogenicity of these mutations is currently lacking [16].

Among the mutations which result in autosomal recessive PD, all characterized by early onset, the ones on the *parkin* gene (*PRKN* or *PARK2*) are the major cause of juvenile PD forms [51,52]. Parkin is an E3-type ubiquitin ligase devoted to the degradation of α-Syn and other substrates. Briefly, mutations in *PRKN* disrupt its E3 activity, resulting in the accumulation of α-Syn and the selective death of neurons in the substantia nigra [53]. It has been observed that a *loss of function* mutation in only one of the two alleles of the *PRKN* gene is sufficient to increase susceptibility for PD or may even give rise to an autosomal dominant inheritance of PD [52,54,55]. Regarding the mutations observed, they have been found in each of the 12 exons of *PRKN* and included point mutations, deletions and duplications [56,57,58,59]. Patients with *PRKN* mutations display the typical PD motor symptoms and present a sustained response to L-dopa; they usually show slow-course dystonia, DA neuron loss in the SNc with sporadic LB and absence of non-motor symptoms. Disease progression is generally slow [60,61].

Mutations in the *PTEN-induced kinase 1* (*PINK1* or *PARK6*) gene also induce features similar to those due to parkin mutations, with typical early-onset motor symptoms, slow progression and lack of non-motor symptoms [62,63]. PINK1 is a mitochondrial serine-threonine kinase responsible for parkin translocation in impaired mitochondria. PINK exerts a neuroprotective role against mitochondrial dysfunction and apoptosis by regulating the specific elimination of dysfunctional or superfluous mitochondria via selective autophagy, thus fine-tuning the mitochondrial network and preserving energy metabolism [64]. Point mutations, frameshift mutations and truncating mutations have been reported throughout the gene, and it has been hypothesized that they may increase the susceptibility to reactive oxygen species and other kinds of stresses, thereby leading to PD [65]. Interestingly, PINK1 and the above-mentioned parkin protein cooperate in an axis that has a key role in the clearance of damaged mitochondria in DA neurons, and it is well-known that a deficiency in this pathway is causative for early-onset PD [66].

*Daisuke Junko-1* (*DJ-1* or *PARK7*) encodes for a transcriptional regulator that protects mitochondria from oxidative stress by increasing the expression of two mitochondrial Uncoupling Proteins (UCP4 and UCP5), thereby decreasing mitochondrial membrane potential and leading to the suppression of ROS production, thus optimizing a number of mitochondrial functions and favoring neuronal cell survival [67,68,69]. Once oxidized, DJ-1 acts as a chaperone for alpha-synuclein, thereby preventing its fibrillation and aggregation [70,71]. The mutations reported for this gene are missense mutations, whole exon deletions, frameshift mutations and a splice site mutation found in either a homozygous or compound heterozygous state [72,73,74,75]. It has been suggested that the mutational state of *DJ-1* might be a good biomarker for PD given the high protein levels found in the cerebrospinal fluid of individuals in the earlier stages of the disease [76].

Obviously, the above-mentioned alternative genes may play substantial roles, just like genes supporting α-Syn, and further studies will lead to the full elucidation of both their contribution in PD and eventual targeting considerations.

### 2.2. Parkinson’s Epigenetics

Epigenetic alterations have been shown to represent a potential linkage between the above-described genetic factors and the environmental conditions at the basis of PD [77]. Along this line, it was found that peripheral blood leukocytes from PD patients display reduced methylation levels in CpG-2 sites of the *SNCA* gene promoter in comparison to controls [78], suggesting that DNA methylation levels might be a potential PD biomarker. Another study of the methylation states of SNCA and its regulatory elements has shown that SNCA expression is upregulated upon methylation-mediated inhibition of a CpG island embedded in intron 1 of the gene; indeed, the putamen and cortex of PD patients exhibited a significant hypo-methylation pattern in this region [79,80].

Additionally, a reduction in the nuclear levels of DNMT1 was reported in both postmortem brain tissue from PD patients and brains from transgenic mice overexpressing α-Syn, this alteration resulting in a global hypomethylation of CpG islands, including the one upstream of SNCA gene [81]. Despite the overall reduction of DNA methylation found in PD, always correlating with high levels of α-Syn and DNMT1 sequestering outside the nucleus, there is not yet a broad consensus on the hypothesis of global changes in DNA methylation occurring in PD.

Instead, there is a general consensus on the role of an imbalance in histone acetylation dynamics in PD: first of all, a number of studies support the idea of a key role of global H3K27 acetylation state in PD, acting through both the regulation of PD-associated α-Syn and the modulation of HDAC activity. Indeed, HDAC inhibition itself can deteriorate DA neuronal function and upregulate SNCA expression [82].

A recent study performed on isolated dopaminergic neurons from PD patients derived brain tissue revealed increased acetylation levels of histones H2A, H3 and H4 in comparison to control individuals. The presence of these histone modifications suggests a significant role of chromatin remodeling in the pathogenesis of PD [83].

It has been hypothesized that epigenetics of inflammation may have a relevant role in PD-related neuronal dynamics by increasing global histone acetylation. Indeed, it has been found that the methylation status of the TNF promoter is drastically reduced in the substantia nigra of PD patients, suggesting that overexpression of TNF can trigger inflammatory cascades able to affect the dopaminergic neurons [84].

#### Epigenetic Alterations in Parkinson’s: The Critical Role of miRNAs

Interestingly, several studies have demonstrated that microRNAs (miRNAs), the short non-coding RNA molecules able to negatively regulate gene expression, may also exert a key role in PD pathogenesis [85,86]. Indeed, different miRNAs have been shown to directly downregulate the *SNCA* gene, their aberrant expression being causative for α-Syn deposition and neuronal cell death [87,88,89]. Accordingly, as described in detail below, reduced levels of these miRNAs have been recently recognized as potential diagnostic biomarkers in PD [90].

A depletion of miR-7, a miRNA responsible for downregulating *SNCA* gene expression, has been found in the brains of PD patients, especially in those regions related to disease neuropathology, such as the substantia nigra, in correspondence with α-Syn accumulation and neuron loss [91]. It has been suggested that a similar role might be played by miR-153, which recognizes sequences in 3′-UTR region of *SNCA* [92]. On the other hand, it has been demonstrated in in vitro models of PD that an increased expression of miR-7 and miR-153 may exert a neuroprotective action over dopaminergic neurons [87,88,89].

miR-34b/c downregulates *SNCA* by targeting a specific 3′-UTR sequence, its abnormal expression affecting α-Syn deposition in PD brain tissue. Observations have been made in PD patients’ brains of a miR-34b/c depletion within the amygdala, frontal cortex, substantia nigra and cerebellum [93].

Experiments performed on SH-SY5Y cells also demonstrated that the depletion of miR-34b/c leads to a reduction in *parkin* and *DJ-1* expression by a not fully elucidated indirect mechanism [93].

miR-133b is specifically expressed in the midbrain, where it regulates both the maturation and function of midbrain DA neurons in a negative circuit that includes the paired-like homeodomain transcription factor Pitx3. Indeed, miR-133b functions within a feedback loop in which Pitx3 specifically induces miR-133b transcription and Pitx3 activity is post-transcriptionally downregulated by miR-133b. Interestingly, Parkinson’s disease midbrain tissue characterized by a massive loss of DA neurons also displays a deficiency in this miRNA [94].

miR-124 is the most abundantly expressed miRNA in neurons [86], in which it regulates synapse morphology, neurotransmission, inflammation, autophagy and mitochondrial function [95]. Interestingly, the bioinformatics exploitation of miRecords, a microRNA-target interactions database, as well as experiments performed on HEK-293S cells, have demonstrated that a plethora of targets of miR-124 are dysregulated in PD [96,97], thus suggesting that a dysfunctional miR-124 downregulation exerts a pivotal role in PD pathogenesis.

In spite of the miRNAs found to be downregulated in PD, other ones, instead, were found to be upregulated, potentially enhancing some aspects of PD symptomatology.

Along this line, the upregulation of some exosomal miRNAs may stimulate many aspects of PD pathogenesis, such as protein aggregation, inflammation and autophagy [98]. Among them, miR-4639-5p displays an upregulated expression in PD cells, where it negatively regulates the post-transcription levels of DJ-1, thus inducing massive oxidative stress and, consequently, neuronal death [99]. Additionally, exosomal miR-137 is upregulated in PD, maximizing its negative regulation of the Oxidation Resistance 1 (OXR1), thus causing neuronal oxidative stress in patients [100].

We can assume that the unbalanced miRNA levels, beyond representing a PD diagnostic marker, may also be exploited as a target for innovative treatment strategies (Table 1).

## 3. Exosome/miRNA Network: Not Only a Role in the Pathogenesis of PD but Also a Clinical Potential for Its Treatment

Currently, there is no cure for PD, and treatments are only palliatives directed to relieve the motor symptoms, inducing severe side effects over time [101]. Among them, Levodopa, the first effective drug for Parkinson’s disease, is still the most used treatment. Levodopa is the immediate precursor to dopamine and allows the depleted number of dopaminergic neurons to produce more dopamine, thus alleviating symptoms. Unfortunately, it has a plethora of side effects, including nausea, somnolence, hallucinations, dystonia and dyskinesia. Other pharmacological treatments are the administration of dopamine agonists, able to stimulate dopaminergic receptors in the central nervous system, and the use of catechol-O-methyl transferase inhibitors and monoamine oxidase aldehyde dehydrogenase B (MAO-B) inhibitors, devoted to inhibition of enzymes involved in the breakdown of levodopa and dopamine, all of them inducing similar side effects to the ones induced by L-Dopa [102].

From here, there is an urgency to discover new drugs able not only to alter the disease progression by alleviating symptoms but also act on the molecular mechanisms at the basis of them. However, the main obstacle in the development of new drug treatments for PD and neurodegenerative diseases, in general, is to cross the blood–brain barrier (BBB) [103]. It is worth noting that exosomes, which are secreted vesicles produced in the endosomal compartment and able to shuttle genetic and protein constituents between cells [104,105], can easily permeate the BBB, suggesting that they may be potential vehicles for drug delivery to the brain [106,107], as will be better explained in the next paragraph. Interestingly, exosomes have demonstrated to own the capability of transmitting miRNAs across brain regions and between cells [108,109,110]. Exosomes in PD, despite their function in mediating intercellular α-syn transmission, also contain a pool of detrimental miRNAs, such as the above-described miR-4639-5p and miR-137 [98]. However, exosomes, due to their bioavailability and ability to cross the blood–brain barrier, may also hold strong therapeutical potential as drug carriers [111,112] of those miRNAs, which are, instead, beneficial for PD treatment.

### 3.1. Exosomes in PD Treatment

To date, it appears that multiple observations indicate that modified exosomes, separated from different cell types, are able to target specific regions of the brain and determine types of neurons, opening a promising scenario for the treatment of PD and other neurodegenerative pathologies [113]. Indeed, it was demonstrated that exosomes, as paracrine factors [114], contain a large number of miRNAs, DNA fragments, proteins and other bioactive molecules, able to shuttle between cells and modify the physiological functions of cells [115]. Most studies have focused on exosomes derived from MSCs [116,117]. In this context, many types of stem cells, such as bone marrow mesenchymal stem cells (BM-MSCs), human embryonic mesenchymal stem cells (hES-MSCs) and induced pluripotent stem cells-derived neural progenitor cells (iPSC-NPCs), have been reported to protect neurons from ischemic stroke-induced death [118,119]. Currently, clinical trials using MSCs for Alzheimer’s disease treatment are ongoing throughout the world [120].

Regarding BM-MSC-derived exosomes, they contain some miRNAs, including miR-146a, miR-133b and miR-21, able to improve neuronal plasticity and promote cell survival [121,122]. Furthermore, experiments performed on an AD mouse model showed that the exosomal transfer of miR-146a secreted from BM-MSCs can be taken up into astrocytes, where it can decrease the levels of NF-κB, thus restoring astrocytic function. This study indicated that the exosomal transfer of miR-146a is involved in the correction of cognitive impairment in an AD mouse model [123].

In regard to iPSC-NPC-derived exosomes, instead, it seems that they may promote neuritic outgrowth with a mechanism not yet fully identified [124].

In addition to their function as paracrine factors, exosomes can be also loaded with bioactive molecules, such as therapeutic compounds and RNAs, and provide specific surface-expressed elements in their membranes that encourage specific delivery to the appropriate cells in the brain.

In this context, a recent study by Qu and colleagues has demonstrated that human exosomes, isolated from the blood and loaded with a saturated dopamine solution, were able to cross the BBB and deliver dopamine into the brain via an interaction between transferrin and transferrin receptors. These dopamine-loaded exosomes have also shown good therapeutic efficacy and low toxicity in vivo in a mouse model and also showed less toxicity than free dopamine by intravenously systemic administration [125].

It has also been shown that an exosome delivery system loaded with the antioxidant catalase has the capability to reach neurons and release the catalase in situ, thus ameliorating neural inflammation and increasing neuronal survival in PD models both in vitro and in vivo [126]. Along this line, a set of EXOsomal transfer into cell (EXOtic) devices was reported to enable the efficient production of designer exosomes in engineered mammalian cells. These genetically encoded devices are able to enhance, in exosome-producing cells, both exosome production and specific mRNA packaging. Interestingly, these devices allow highly efficient delivery of the mRNA into the cytosol of target cells by taking advantage of the presence, on the exosomes’ surface, of a targeting module, RVG-Lamp2b, able to target exosomes to the brain by binding to the nicotinic acetylcholine receptor (CHRNA7) [127]. This system has been demonstrated to work well in a living mouse model of PD; indeed, the implantation in the brain of engineered cells, able to produce exosome-containing packaged catalase mRNA, attenuated both neurotoxicity and neuroinflammation [128].

Other studies have demonstrated therapeutic potential for PD treatment by using exosomes carrying small interfering RNAs (siRNAs): *SNCA*-siRNA-exosomes containing *SNCA*-siRNA have been shown to reduce both mRNA transcription and translation of α-syn in the brain of the S129D α-syn transgenic mice [129].

Additionally, short hairpin RNAs (shRNAs) have demonstrated their efficacy when delivered to the brain using exosomes as transport vehicles. Izco and colleagues, indeed, have designed shRNA minicircles which, when delivered by Rabies Viral Glycoprotein (RVG)-exosomes to target *SNCA* mRNA in a PD mouse model, not only reduced synuclein aggregation, but also decreased dopaminergic neuron death [130].

### 3.2. Mesenchymal Stem Cell (MSC)-Derived Exosomes as Therapeutic Vehicles for miRNA Delivery in PD Treatment

PD is characterized by an imbalance of the miRNA pool, observed either as an upregulation of those detrimental exosomal miRNAs involved in protein aggregation, inflammation and oxidative stress induction [98,99,100] or as a depletion of those miRNAs responsible for downregulating SNCA gene expression, thus enhancing α-Syn accumulation and disease neuropathology [87,88,89]. This is why, in addition to the above-described exosome-mediated RNA interference (RNAi) strategy, which could be a powerful tool with high clinical potential for treating PD, innovative approaches acting on RNA activation (RNAa) [131], by restoring the physiological miRNAs pool, which is altered in PD, could be another effective and complementary strategy to restore the correct gene expression in PD.

As anticipated above, mammalian cells capable of being implanted and secreting exosomes loaded with drugs have demonstrated a good therapeutic potential to achieve this aim. In particular, Mesenchymal stem cell (MSC)-derived exosomes have yet shown to exert beneficial effects in both cancer and a number of pathologies [132,133], including PD, as demonstrated in the 6-OHDA mouse model of PD, in which MSC-derived exosomes efficiently rescued dopaminergic neurons [134].

Worthy of note, it was also demonstrated that MSC-derived exosomes can carry miRNAs and interact with neuronal cells, as in the case of the above-described miR-133b, expressed in midbrain dopaminergic neurons and involved in the regulation of the production of tyrosine hydroxylase. A study by Xin and colleagues has demonstrated that MSC-derived exosomes carrying miR-133b to neurons and astrocytes can regulate neurite outgrowth by regulating target genes, such as RhoA, that stimulate neurite outgrowth, thereby improving functional recovery after stroke [135].

The activation of NAcht Leucine-rich repeat Protein 3 (NLRP3) inflammasome/pyroptosis and cell division protein kinase 5 (CDK5)-mediated autophagy is known to play an important role in PD, leading to the dopaminergic degeneration and microglial activation. Starting from the bioinformatic prediction that (miR)-188-3p might target both NLRP3 and CDK5, Li and collaborators, taking advantage of AD-MSCs-EXOs (Adipose-derived Mesenchymal Stem Cells-derived exosomes), performed an elegant study showing that the injection of miR-188-3p-enriched exosomes in PD mice both inhibits autophagy and enhances proliferation by binding to CDK5 and NLRP3 [136]. This study demonstrated that treatment with miR-188-3p-enriched exosomes displays a good restoration effect on damaged neurons in the substantia nigra of a PD mouse model, providing the basis for a hypothesis that miR-188-3p-enriched exosomes may become an effective treatment for PD patients [136].

Overall, it is clear that the transfer of miRNAs within MSC-derived exosomes is beneficial to PD cells and animal models and appears to be a promising future strategy for PD treatment (Figure 1).

## 4. Conclusions and Future Perspectives

PD is one of the most common neurodegenerative disorders, characterized by an initial and progressive loss of dopaminergic neurons of the substantia nigra pars compacta via a potentially substantial contribution from protein aggregates, the Lewy bodies, which are mainly composed of α-Synuclein among other factors. PD’s typical symptoms are bradykinesia, muscular rigidity, unstable posture and gait, hypokinetic movement disorder and resting tremor [1,2,3]. Currently, there is no cure for PD, and palliative treatments, such as L-Dopa administration, are only directed to relieve the motor symptoms and induce severe side effects over time [102]. The evidence of epigenetic alterations, such as the dysregulation of different miRNAs that may stimulate many aspects of PD pathogenesis, opened new and promising scenarios for treatments directed against the molecular mechanisms at the basis of PD. Even if, to date, there is no established therapy or ongoing clinical trials using miRNAs, there is an overall consensus on their potential exploitation as a therapeutic strategy for PD treatment [137]. Along this line, modified exosomes, due to their capability of being loaded with miRNAs and of being deliverable to the appropriate location in the brain, overcoming the blood–brain barrier, appear to be an eligible vehicle for miRNA transferring to the brain in a system that has yet demonstrated successful results both in vitro and in vivo [134,135,136].

However, despite the strong potential shown by exosomes as transport vehicles for miRNAs able of modifying gene expression to alter PD prognosis or progression, there are some limitations that need to be overcome.

First of all, the current technology to obtain exosomes needs to be improved in order to have purer exosomes. Indeed, despite the growing interest in exosome properties and capabilities for disease treatment, the technologies for their purification and enrichment are still quite rudimentary. The isolation of exosomes, due to their small size and heterogeneity, is still challenging. Different systems are applied, such as Charge-Based Methods, in which exosomes are manipulated by electrophoresis taking advantage of their strong negative zeta-potential at physiological pH [138]; Label-Based Methods, in which surface markers are used for exosome isolation [139,140]; and Size-Based Methods, in which different size-based principles. Regarding the size range for exosomes, it is expected to be 50–150 nm, and different size-based principles are applicable in order to purify them: filtration, in which membranes with two different cutoff sizes [141] or a series of double-filtration devices are used [142]; Deterministic Lateral Displacement (DLD), consisting of pillar arrays with a gap distance by which particles can be bumped and isolated [143]; Asymmetric Flow Field-Flow Fractionation (AF4), a purification technique which takes advantage of the different size of particles for fractionating and collecting exosomes from tissue culture medium at specific time sections [144]; and, finally, DiElectroPhoresis (DEP) can sort the particles by size through a polarization force produced by a nonuniform electric field [145]. All the above-mentioned techniques are receiving ongoing studies in order to achieve better results [146].

Another reasonable concern arises from the quality control of exosome purity, which is still an open challenge: indeed, there is no candidate protein that is a unique and specific marker for exosomes. To date, the mainly utilized markers include components of Endosomal Sorting Complexes Required for Transport (ESCRT), such as Alix and TSG101; Rab small GTPases; and tetraspanin proteins, including, among others, CD81, CD63 and CD9 [147,148]. The latter is widely used for isolation because of exosomal typical surface proteins. However, they are barely specific and not ubiquitously expressed [149], and for this reason, studies are ongoing in order to obtain better profiling of exosomal surface proteins [150].

Regarding the method for loading miRNA or siRNA into exosomes, the most common, until recently, was electroporation. Even if this strategy has demonstrated efficacy in loading siRNA into purified exosomes [151], unfortunately, transfection of exosomes directly with nucleic acid by this system is not quite efficient since it requires both the separation and purification of exosomes before and after transfection, thus drastically reducing the exosome quantity [152]. Along this line, a new and promising scenario has opened with the EXOtic devices developed by the Kojima group, which enable the specific and highly efficient delivery of mRNA and have been demonstrated to be a good method for loading RNAs into exosomes [128]. This system needs to be further studied for the application of exosomes in clinical practice.

Another limitation of exosomes is that they are usually obtained in small yields [153]. To improve the production efficiency of exosomes, several methods to enhance the total amount of exosomes are under study. Among the others, one strategy is to increase the intracellular calcium levels in order to enhance the formation of extracellular vesicles [154]; another strategy is to induce vesicular formation by cytoskeletal blocking in order to stimulate vesicle production [155]; furthermore, hypoxia [156], thermal stress [157], radiation [158] and pH [159] can also increase exosomes production; and, finally, the addition of liposomes in vitro has shown to increase the yield of exosomes [160]. However, the application of these methods may affect the packaging and delivery efficiency of nucleic acid into exosomes. Therefore, further studies are required to improve the production of exosomes while maintaining their biological activity [152].

Last but not least, exosomes cannot be stored for a long time. Therefore, it is also necessary to improve exosome preservation technology to protect their biological activities and make them suitable for clinical application [161,162,163,164,165]. Again, help may come from synthetic biology-inspired cell-based treatment strategies, such as the above-described EXOtic devices developed by the Kojima group, which are based on implanted designer cells able to produce and secrete therapeutic molecules. Indeed, this system has allowed the constitutive production and delivery of drug-encoding mRNAs inside exosomes, overcoming the problems due to the limited half-life of exosomes administered intravenously [128].

The last open question regards the choice of eligible patients for this kind of treatment. Obviously, innovative strategies able to diagnose PD before the main part of dopaminergic neurons have deteriorated are mandatory for the effectiveness of this therapeutic approach. In this context, the analysis of plasma levels of these miRNAs in people with a family PD history could have the potential to early detect PD onset. Having said that, a possible diagnostic criterion to select conclaimed PD patients for this kind of therapy could be the analysis of miRNAs isolated from their monocytes. Along this line, a recent study has yet to successfully compare miR-124-3p expression levels in monocytes from non-smokers or former smokers to the ones from smokers [166]. This kind of analysis would ideally provide a personalized strategy able to target only the specific miRNA/miRNAs aberrantly regulated in each clinical case.

Overall, it is clear that, even if there are still some challenges to be overcome, this kind of treatment that takes advantage of exosomes for the delivery of therapeutic miRNAs to the brain will be the future for treating PD patients.

## Figures and Tables

**Figure 1 ijms-24-09547-f001:**
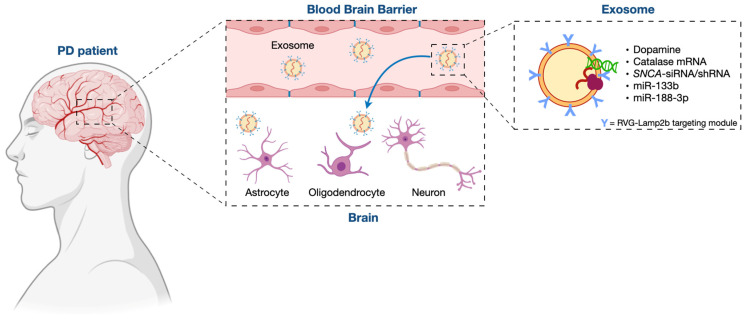
The potential role of miRNA-loaded exosomes for PD treatment. Modified exosomes, loaded with miRNAs or other bioactive molecules, are able to overcome the blood–brain barrier and reach the appropriate location in the brain, thus appearing to be an eligible vehicle for therapeutics transferring to the brain.

**Table 1 ijms-24-09547-t001:** The role of miRNAs in Parkinson’s disease. Dysregulation of different miRNAs may stimulate many aspects of PD pathogenesis. The table shows some of the miRNAs most frequently found to be aberrantly expressed in PD, the target gene which they regulate and the molecular mechanisms that appear to be altered in PD.

**miRNAs downregulated in PD**	**Target genes**	**Molecular mechanisms**
**miR-7**	*SNCA* [91]	miR-7 is responsible for downregulating *SNCA* gene expression, its depletion being associated with α-Syn accumulation and neuron loss
**miR-153**	*SNCA* [92]	miR-153 is responsible for downregulating *SNCA* gene expression, its depletion inducing α-Syn accumulation and neuron loss
**miR-34b/c**	*SNCA*, *Parkin* and *DU-1* [93]	miR-34b/c is responsibl for dowregulating *SNCA* gene expression, its depletion leading to both α-Syn deposition in PD brain tssues and downregulation of *Parkin* and *DJ-1* gene expression
**miR-133b**	*Pitx3* [94]	miR-133b is specifically expressed in midbrain, where it regulates both the maturation and function of midbrain DA nourons, its depletion being associated with a massive lost of DA neurons
**miR-124**	*Calpain 1*, *Bim*, *STAT3*, *Annexin A5*, *MEKK3* [95,96,97]	miR-124 regulates synapse morphology, neurotransmission, infammation, autophagy and mitochondrial function, its depletion being implicated in the core pathophysiologic mechanisms
**miRNAs upregulated in PD**	**Target genes**	**Molecular mechanisms**
**miR-4639-5p**	*DJ-1* [99]	miR-4639-5p negatively regulates the post-transcription levels of DJ-1, its upregulation being responsible for a massive induction of oxidative stress and, consequently. neuronal death
**miR-137**	*OXR1* [100]	miR-137 is involved in the induction of oxidative stress in neurons, its upregulation being involved in a massive induction of oxidative stress and neuronal death

## Data Availability

No new data were created or analyzed in this study. Data sharing is not applicable to this article.

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
