# Peer review of "Parkinson’s Disease: From Genetics and Epigenetics to Treatment, a miRNA-Based Strategy"

_ijms, 2023, doi:10.3390/ijms24119547_

Round 1
Reviewer 1 Report
-
Author Response
We thank reviewer #1 for his positive recommendation.
Reviewer 2 Report
This review, providing a systematic over-view of both the genetic and epigenetic basis of the disease, aims to explore the exosomes/miRNAs network and its clinical potential for PD treatment. It appears comprehensive and reasonable. There are several points need to note:
1. Since your title is related to genetics and epigenetic, however, you need set up a section to deal with epigenetic part you raise.
2. Why not deal with the mitochondrial protein kinase PINK1/Parkin (as E3), which are required for mitochondrial maintenance in dopaminergic (DA) neurons whose degeneration also leads to the development of PD.
3. On 3.2. Mesenchymal stem cell (MSC)-derived exosomes as therapeutic vehicles for miRNA delivery in PD treatment. Better point out, compare and discuss other types, such as: human embryonic stem cells (hES-MSCs); bone-marrow-derived MSCs (BM-MSCs) exosomes?
4. Better discuss and compare the quality control of exosomes (a): what size for these exosomes? (b) Biomarker? Their pros and cons?
5. On table 1 Refs 83-84 need to change into 82-84 for miR-124 target genes column.
Author Response
We thanks the reviewer for the kind revision of our work and for the useful suggestions. Please, find below our point to point answers.
This review, providing a systematic over-view of both the genetic and epigenetic basis of the disease, aims to explore the exosomes/miRNAs network and its clinical potential for PD treatment. It appears comprehensive and reasonable. There are several points need to note:
1. Since your title is related to genetics and epigenetic, however, you need set up a section to deal with epigenetic part you raise.
According to the reviewer suggestion we added this section, please see paragraph 2.2.
2. Why not deal with the mitochondrial protein kinase PINK1/Parkin (as E3), which are required for mitochondrial maintenance in dopaminergic (DA) neurons whose degeneration also leads to the development of PD.
According to the reviewer suggestion we mentioned the PINK1/Parkin axis, please see paragraph 2.1.
3. On 3.2. Mesenchymal stem cell (MSC)-derived exosomes as therapeutic vehicles for miRNA delivery in PD treatment. Better point out, compare and discuss other types, such as: human embryonic stem cells (hES-MSCs); bone-marrow-derived MSCs (BM-MSCs) exosomes?
According to the reviewer suggestion we extensively discussed this point, please see paragraph 3.1.
4. Better discuss and compare the quality control of exosomes (a): what size for these exosomes? (b) Biomarker? Their pros and cons?
According to the reviewer suggestion we extensively discussed these aspects, please see chapter 4.
5. On table 1 Refs 83-84 need to change into 82-84 for miR-124 target genes column.
We fixed this point.
Reviewer 3 Report
I have attached a general Review file about this article that would have been significantly aided had the original article available had line numbers.

I have gone through and made extensive suggestions for correcting odd-seeming phrasing throughout the document.
Author Response
We are really grateful to the reviewer for the efforts he spent in suggesting improvements for this manuscript. Also, we appreciate his deep knowledge of the literature that has been very useful to better contextualize our work.
1) The goals of this paper seem clear, and the concept is worthy of addressing with substantial scholarly effort tracking several molecular elements. It appears that there are statements that raise the α-Synuclein (αSyn) and related Lewy bodies to a level of general and confusingly-almost-deemed sole causative factor across all forms of Parkinsonism, which doesn’t seem justified to begin with. This reviewer would like those statements tempered to “contributors to deterioration in Parkinsonism,” rather than the solely “due to the accumulation of protein aggregates...”. There is a current trend that is infiltrating even textbooks at earlier levels which seems to raise these factors up into a status of being considered the sole culprits in Parkinson’s disease regardless of how Parkinson’s disease is established. While there may be an association with these Lewy bodies sought as a confirmation of Parkinsonism in postmortem brains, this is not sufficient to raise to the level of almost sole causation as is done. There seems to need to be some tempering of these statements. This reviewer understands how this happens, as there are strong desires to justify and elevate models to the level of predominance. Yet this has happened with Alzheimer’s as well with the beta amyloid plaques and/or the tau molecules that people strongly advocate as the main cause and are often considered to be only a part of a larger equation. There are several references that need to be acknowledged along the way to address this issue:
- For years, there has been a clear dissociation in driving mechanisms between the familial and thus more hereditary versions of Parkinsonism such as the ones driven by various LRRK2, PARK7, PINK1, PRKN, or SNCA genes (Medline Plus speaks of about 15% of cases connected to these classic genetic and “familial” bases: https://medlineplus.gov/genetics/condition/parkinson-disease/#causes). While the authors of our addressed paper do mention several of these genes throughout the document as being potentially intermixed even in the more sporadic or idiopathic cases, the larger numbers of patients clearly have a wider variety of different factors at work. In many cases, there are mitochondrial effects that are also partially mentioned by the authors, which may not require or derive from the increases in αSyn. A wide variety of other rationales for oxidative processes have been referenced over the years as other potential causative factors that might derive via mechanisms alternative to αSyn.
2. Explorations of brains of parkinsonian patients after the fact, when not exclusively chosen with the criterion of having specific increases in αSyn and Lewy bodies, have demonstrated wider varieties of contribution from these factors
Calne, DB & Mizuno, Y. (2004) The neuromythology of Parkinson's disease. Parkinsonism & Related Disorders, 10(5), 319-322. https://doi.org/10.1016/j.parkreldis.2004.03.006
Gasser, T (1998) Genetics of Parkinson's disease. Clinical Genetics, 54, 259-265. https://doi.org/10.1034/j.1399- 0004.1998.5440401.x
3. Other respected reviews of causative factors for parkinsonism have also presented αSyn and related Lewy bodies as one of many contributing factors to the deterioration in Parkinson’s disease across expressed versions
Devine, MJ, Gwinn, K, Singleton, A, and Hardy, J. (2011) Parkinson's disease and alpha-synuclein. Movement Disorders 26(12), 2160-2168. https://doi.org/10.1002/mds.23948
4. One paper, written by Braak et.al., which does a nice job showing how progress in (αSyn) and related Lewy body distribution follows a specific timecourse and spatial pattern across the brain has often been used as a basis to advocate for this generality of αSyn-focused and dependent parkinsonism across all expressions, is not appropriate to advocate this point. Braak’s study specifically sought out postmortem brains that were riddled with Lewy bodies. That study was not about developing αSyn and Lewy bodies as the root, primary, or sole cause of deterioration of dopaminergic neurons in Parkinson’s disease. This article did claim that, “A prerequisite for the post-mortem diagnosis of both the presymptomatic and symptomatic phases of the pathological process underlying PD is evidence of specific inclusion bodies, which develop as spindle- or thread-like Lewy neurites (LNs) in cellular processes, and in the form of globular Lewy bodies (LBs) in neuronal perikarya." Yet that statement implies that this status is often used as a further confirmation of parkinsonian status on top of other things, and neither the article as a whole, nor the world of current evidence, supports the idea that the αSyn and Lewy bodies are the sole, or even the main reason for all the deterioration that happens in Parkinsonism.
Heiko Braak, Kelly Del Tredici, Udo Rüb, Rob A.I de Vos, Ernst N.H Jansen Steur, Eva Braak. (2003) Staging of brain pathology related to sporadic Parkinson's disease. Neurobiology of Aging. 24(2), 197-211. https://doi.org/10.1016/S0197-4580(02)00065-9
5. Most importantly, the causative connection between (αSyn) and related Lewy bodies in terms of deterioration being such a sure thing as indicated was directly addressed with several caveats in this paper which maintains a similar perspective as this reviewer (not the author of this paper), which is that such aspects are a part of a wider picture and variably involved in the deteriorative process:
Schulz-Schaeffer, W. (2015) Is cell death primary or secondary in the pathophysiology of idiopathic Parkinson's disease? Biomolecules, 5(3), 1467-1479. https://doi.org/10.3390/biom5031467
Having said all this, this reviewer would first like to advocate a more appropriate phrasing of three key places in the current document where the sentences include “...due to the accumulation of protein aggregates, the Lewy bodies, mainly composed of α-Synuclein.” Advocated change in the sentence to this:
‘...via a potentially substantial contribution from protein aggregates, the Lewy bodies, mainly composed of α-Synuclein among other factors.’
This adjustment should be added to the abstract where the red statement shows up on the first page, as well as where the sentence is in the form of:
“Both sporadic and familial forms of PD are characterized by an initial and progressive loss of dopaminergic neurons (DA) of the substantia nigra pars compacta (SNc) due to the accumulation of protein aggregates, called Lewy bodies (LB), mainly composed of α-Synuclein (α-Syn).”
A similar adjustment should be made to the related statement in the conclusion (beginning of section 4 on page 7 of 19).
Alternative genes are even mentioned in your own review (fourth paragraph under section 2.1) which seem to be casually mentioned and then dismissed as essentially irrelevant given that we don’t know much about them. This reviewer advocates making it clear that these alternative genes may well play substantial roles, just like genes supporting α-Syn, and also may lead to avenues of adjustment with future targeting considerations with a technique tested on what we seem to have more data about (α-Syn). Further, the connection with respect to reactive oxygen species mentioned at the end of the first full paragraph on page 3 of 19 (starting with “Mutations in PTEN-induced kinase 1...”) was directly mentioned with doesn’t necessarily harbor a direct connection to α-Syn. In fact, these are more clearly straight mitochondrial issues.
The point is, there are likely a number of genes that are variously affected within both familial forms referenced by this reviewer previously (representing a small proportion of overall Parkinson’s disease patients), and the larger sporadic or idiopathic form where such specific genetic mutations do not stand out as the main component but may contributed even substantially as a part of the equation.
We rephrased the sentences as suggested by the reviewer and we also introduces all the references that he indicated. Please see chapter 1 and paragraph 2.1.
2) The next issue is a clarity issue that may derive in part from a lack of control one has over the contents of exosomes or essential factors that might need to be included when one yields benefits from them. On page 6 of 19, in the second paragraph under heading 3.2, mesenchymal stem cell derived exosomes were referenced as having “beneficial effects in both cancer and a number of pathologies, including PD, ... etc.” It seems completely unclear what components are contributing to this cause since these exosomes appear to simply be beneficial solely because they derived from a powerful precursor cell, potentially containing multiple elements having nothing to do with the main concern of this article (that clearly being αSyn). It seems that some elaboration of what might be known about these contributions might be delivered, or at the very least the conclusion might be reached that general benefits in a context might require significant investigation before knowing the nature or key ingredient relevant to the effect.
According to reviewer suggestion and on the basis of the existing literature we better argument the issue raised by the reviewer. Please, see chapter 3.
3) The next issue rears the question of targeting capacity of these elements such as exosomes or miRNAs to be delivered in a targeted fashion is very loosely discussed using a phrase like “functionalized on the surface for specific delivery...” [a phrase by the way that seems to make no sense on its own, as this reviewer is not aware of a word like functionalized being used in this context, correction suggestion made in next section]. Here it seems like a greater degree of discussion is warranted for the targeting capacity of these delivery mechanisms advocated and the tendency for them to land in desired versus undesired areas (off-target delivery) should be considered. In several cases, a wide and undirected delivery of gene- manipulating components might be considered clinically sound – such as in the case of controlling the expression levels of the mutated huntingtin gene in Huntington’s disease. However, the genes discussed in this article seem to have multiple roles, and even if the suppression may not reach 100% (extent of suppression also not addressed), it remains of concern for a wide influence of such factors being delivered into potentially alternative populations with very different roles, leading to very unfortunate consequences. You mention in your conclusion the following sentence: “Along this line, modified exosomes, due to their capability of being loaded with miRNAs, and of being deliverable to the appropriate location in the brain, overcoming the blood brain barrier, appear to be an eligible device....” This reviewer would recommend both reworking that sentence and providing more support for the “being deliverable to the appropriate location in the brain” since the sentence is both run-on and lacks sufficient support in your review. Perhaps also, in your Figure 2, the components placed and colored blue as if they are binding molecules on the exosome depicted, might be discussed a bit more.
According to reviewer suggestions we i) modified Figure 1 by naming component colored in blue and ii) better explained how exosomes can be delivered in brain. Please see paragraph 3.1.
4) Table 1 and the various miRNAs discussed, and their apparent roles was very useful, and this reviewer was thankful for its inclusion. However, some comprehension pain came from trying to juxtapose the downregulation of miaR-153 and miR-34b/c with the upregulation of miR-4369-5p in terms of the influence these two PD-related alterations have on DJ-1 and perhaps subsequently α-Syn. The logic this reviewer attempted to apply was that the removal of components that downregulated genes potentially supporting α-Syn increases, α-Syn would increase. Removal of components meant to “negatively [regulate] the post-transcription levels of DJ-1” might mean that miR4369-5p is meant to help keep a cap on the levels of DJ-1 which is already challenged by the removal of an element meant to control the original expression of the gene. Both this connection, and the relationship between DJ-1 and α-Syn seem to need some reinforcing to complete the circle.
We completely agree with the reviewer. It was our fault and the table has been reworded accordingly. Moreover we better explained the different involvements of the indicated miRNAs in the text. Please, see chapter 2.
5) The fourth paragraph under section 3.2 is quite difficult to follow as written. I will be making a separate suggestion in the next section (minor deficiencies needing correction) of a phrasing toward the end. However, the general point is lost in the delivery. It seems to this reviewer that evidence of NLRP3 inflammasome/pyroptosis and CDK-mediated autophagy should be explained first. Following that, the potential for their downregulation via miR-188-3p should be raised with a focus on the substantia nigra. For purposes of delivery, it might be helpful to indicate whether this was directly injected into the SPc or if it was given systemically. Something needs to be done to hone the delivery of that point.
According to the reviewer suggestion we rewrote the paragraph.
6) A general question that arises that deserves some commentary is this: Which patients would be candidates for this sort of treatment?” Generally speaking, most PD patients don’t know much about their disease status until over about 70% of their dopaminergic neurons have deteriorated. There really are very few efforts under way to capture an earlier diagnosis outside of some clinically directed genetic/familial aspect that might be gathered, yet those are often not sought unless there is some apriori clinical rationale. This preventative mechanism seems to make diminished sense to simply be given to all people starting at age 60. Might there also be diagnostic criteria that promote some PD patients as good candidates for this therapy? The end goal thus remains unclear.
The reviewer pointed out a key issue. In the last part of the chapter 4 we discussed possibile strategies for identify patients that could benefit from this therapeutic approach.
7) Individual Minor Deficiencies Needing Correction (table)
We fixed the points.
Round 2
Reviewer 2 Report
The present version has been sufficiently improved to warrant publication in IJMS.
There is no further comment!
Author Response
Thank for your prompt revision.
Best Regards
Reviewer 3 Report
This reviewer thanks the editorial team for producing a line-numbered version for this go-round. This reviewer found the revised version VASTLY improved and appreciates the attention given to the points raised. It is necessary at this point to admit that it is possible to have Parkinson's disease without obvious signs of alpha-synuclein Lewy bodies, though there is high involvement overall. Generally-speaking this revision is highly acceptable, though this reviewer would like to point out one remaining English issue (see next commentary).
Line 262:
The sentence reads "Exosomes in PD, despite their function in mediating intercellular α-syn transmission, also contain pool of detrimental miRNAs..."
The above-highlighted word "pool" either needs to be changed to "a pool" or to "pools" to make this sentence stand in English. If agreed, this change might simply be made by the copy editors without any more extensive responses.
Author Response
Thank for your prompt revision. We fixed the point you raised.
Best Regards